# Genome-Wide Searching Single Nucleotide-Polymorphisms (SNPs) and SNPs-Targeting a Multiplex Primer for Identification of Common *Salmonella* Serotypes

**DOI:** 10.3390/pathogens11101075

**Published:** 2022-09-21

**Authors:** Md-Mafizur Rahman, Sang-Jin Lim, Yung-Chul Park

**Affiliations:** 1Division of Forest Science, Kangwon National University, Chuncheon 24341, Korea; 2Department of Biotechnology and Genetic Engineering, Faculty of Biological Science, Islamic University, Kushtia 7001, Bangladesh; 3Institute of Forest Science, Kangwon National University, Chuncheon 24341, Korea

**Keywords:** single nucleotide polymorphisms (SNPs), *Salmonella enterica*, SNP-multiplex primer, prevalence, genes

## Abstract

A rapid and high-quality single-nucleotide polymorphisms (SNPs)-based method was developed to improve detection and reduce salmonellosis burden. In this study, whole-genome sequence (WGS) was used to investigate SNPs, the most common genetic marker for identifying bacteria. SNP-sites encompassing 15 sets of primers (666–863 bp) were selected and used to amplify the target *Salmonella* serovar strains, and the amplified products were sequenced. The prevalent *Salmonella enterica* subspecies enterica serovars, including Typhimurium; Enteritidis, Agona, enterica, Typhi, and Abony, were amplified and sequenced. The amplified sequences of six *Salmonella* serovars with 15 sets of SNP-sites encompassing primers were aligned, explored SNPs, and SNPs-carrying primers (23 sets) were designed to develop a multiplex PCR marker (m-PCR). Each primer exists in at least two SNPs bases at the 3′ end of each primer, such as one was wild, and another was a mismatched base by transition or transversion mutation. Thus, twenty-three sets of SNP primers (242–670 bp), including 13 genes (*SBG*, *dedA*, *yacG*, *mrcB*, *mesJ*, *metN*, *rihA/B*, *modA*, *hutG*, *yehX*, *ybiY*, *moeB*, and *sopA*), were developed for PCR confirmation of target *Salmonella* serovar strains. Finally, the SNPs in four genes, including *fliA* gene (*S.* Enteritidis), *modA* (*S*. Agona and *S*. *enterica*), *sopA* (*S.* Abony), and *mrcB* (*S.* Typhimurium and *S*. Typhi), were used for detection markers of six target *Salmonella* serotypes. We developed an m-PCR primer set in which *Salmonella* serovars were detected in a single reaction. Nevertheless, m-PCR was validated with 21 *Salmonella* isolates (at least one isolate was taken from one positive animal fecal, and *n* = 6 reference *Salmonella* strains) and non-*Salmonella* bacteria isolates. The SNP-based m-PCR method would identify prevalent *Salmonella* serotypes, minimize the infection, and control outbreaks.

## 1. Introduction

Single nucleotide polymorphisms (SNPs), a common allelic variation in all organisms, exists in whole-genome sequences of *Salmonella*. Identification of SNPs by comparing sequence data from target *Salmonella* serovars with a reference genome sequence and varying nucleotides are created an SNP matrix [1,2]. SNPs are highly informative and stable markers that can be used to efficiently detect, investigate outbreaks, and reveal evolutionary analysis of similar bacterial groups [1,2]. The advent of whole-genome sequence (WGS) has improved the ability to investigate outbreaks by exploiting SNPs that may vary among isolates [3].

*Salmonella enterica* serovars Typhi (typhoidal) are restricted to humans, whereas nontyphoidal *Salmonella* (NTS) serovars are generalist pathogens with different hosts (wild and domestic animals), which express as asymptomatic carriers [4]. NTS *Salmonella enterica,* i.e., Enteritidis, and Typhimurium are the two major zoonotic serovars that cause illness, including diarrhea, gastroenteritis, septicemia, and other clinical syndromes [5,6]. *S.* Typhimurium is a broad host range of serovar, including wild and domestic animals, which are recognized as a leading NTS infectious agent (approximately a quarter of infections out of total NTS infections), and *S.* Enteritidis is recognized as the second infectious agent [7]. *S.* Agona infections commonly occur in humans by consuming contaminated animal food [8,9]. The ultimate consequence is that two billion humans are annually suffering from *Salmonella* gastroenteritis, leading to over 3 million deaths globally [10]. The prevalent *Salmonella* is widely found in diverse sources, including environment, animal-originated foods, water, and animals, specifically wild animals, including wild birds [11,12,13], wild pigs [14], poultry and their eggs [15], and cattle [16]. It can be disseminated to humans via ingestion of contaminated food of animal origins, including pork, chicken, eggs, beef, and milk [17,18,19].

Serotyping methods can be used for the characterization of over 2800 distinct serovars of *Salmonella enterica,* which are costly, time-consuming, labor-intensive, and insensitive [20]. So, a rapid and accurate detection method is essential for identifying prevalent and high-risk *Salmonella* serovars. In addition, serotype information is needed for animal-originated foods safety and the public health burden of salmonellosis. Nevertheless, serotypes may provide important epidemiological data, as well as specific virulence characteristics with specific contamination sources [20,21]. Therefore, public health and regulatory agencies need a rapid, highly accurate, and discriminatory SNP-based method to detect serotypes and outbreaks, and to link illness cases to the incidence investigations [21,22,23]. Thus, to achieve serovar-specific detection of six *Salmonella enterica* serovar strains in a single multiplex PCR amplification, we implement the SNP-existing genes of WGS *Salmonella*. Nevertheless, several researchers have utilized mutation sites in different genes to provide a molecular target to establish a method due to its excellent specificity [24,25,26,27].

The current research aims to identify novel sensitive, and reliable serovar-specific targets and develop an m-PCR method for *Salmonella* serovars to facilitate timely prevention and treatment. Several developed detection methods are conducted for *Salmonella* serovars Enteritidis [28], and Typhimurium [29], since they top the list of the most prevalent serotypes [30,31,32]. *S*. Enteritidis is the largest (between 40% and 60% of human illness) pathogens of *Salmonella* infection and disease outbreaks in humans globally [30,33,34]. However, a few SNP-based molecular identification studies are conducted on multiple serovars of *Salmonella* in a single PCR reaction [35]. In research, a *Salmonella*-serovar-specific multiplex marker was developed using SNPs in gene fragments (flagellin gene, *fljB*, DNA gyrase, *gyrB*, and putative stress regulatory gene, *ycfQ*) and evaluated for serotype-specific subtyping of *Salmonella enterica* isolates. So far, to our knowledge, a few studies with SNP-based multiplex PCR markers were developed, and detection in which widely prevalent *S. enterica* serovars (typhoidal and non-typhoidal) were detected in a single reaction. Therefore, we conducted a molecular study of six *Salmonella enterica* subsp. enterica serovars (Typhimurium, Typhi, Enteritidis, *S.* Abony, Agona, and *S. enterica*) identification with a developed m-PCR marker.

## 2. Results

### 2.1. Acquired Salmonella WGS from GenBank

Three *Salmonella bongori* WGS sequences were downloaded and aligned for investigating SNP sites. Among the three, we regarded one reference (NC-015761) and two comparing strains (NC_021870, NZ_CP006692). The accession number, SNP positions, length of reference WGS, and compared *S. bongori* strains are provided in Table 1 and Appendix A.

### 2.2. Searching SNP Sites from NGS of Salmonella Genome Alignment and Design Primers Based on SNP Sites

The *Salmonella* genome sequences were obtained from NCBI and compared to the reference sequences. We found 140 SNPs on the alignment of *S. bongori* serovar genomes (Appendix A). Within the SNPs, we detected a number of SNPs (functional, high-quality SNPs based on nonsynonymous and synonymous mutation) on the aligned WGS of *Salmonella* using Bioinformatics software. We thus selected 15 sets of SNP sites surrounding primers based on 13 genes of the *Salmonella* genome (Table 1). Detailed information on SNPs with the position of ambiguous codes and amino acids in the respective genes of a reference *Salmonella* strain (NCTC 12419) is provided in Table 1.

### 2.3. Amplified Target Salmonella Serovars with Newly Designed Encompassing-Primer Sets

We developed 15 sets of SNP markers (666–863 bp) from the NGS data analysis of *Salmonella* genomes (Table 1). The amplified PCR products with PCR results (amplicon length) are marked in Appendix A, which describes the amplification of PCR with 15 primer pairs. Among 15 primers, the six primers (09-, 12-, 15-, 16-, 18-, and 24-Sbon) were amplified with desire genes of target *Salmonella*, whereas six primers (04-, 06-, 11-, 14-, 19-, 21-, 22-Sbon) were not amplified with desire genes of *Salmonella* (Appendix A). Five primer pairs (1-, 9-, 13-, 14- and 24-Sbon) were amplified in the first PCR, and seven primer sets (11-, 12-, 15-, 16-, 18-, 19-, 21-Sbon) were amplified in the second PCR amplification. However, the three (4-, 6, and 22-Sbon) primer sets have not produced any band in both amplification (first and second PCR) (Appendix A).

### 2.4. Justification of SNP Sites on Target Salmonella Gene Sequences and Design with Serotype-Specific SNP Primers

Among the 15 primers, 12 primers were amplified at the target band properly during the first (five primers) and second PCR (seven primers), and the rest of the three primers failed to produce any band (Appendix A). For instance, the forward primer (12-Sbon-F): 5′-ATTGGCACGCTGTCAGCT-3′ and the reverse primer (12-Sbn-R): 5′-TGCCGGTAAAAGCACGCT-3′ were used to amplify the target band (681 bp) and desired SNP positions of reference *Salmonella* strain (270836: G of *S. bongori*, NC-015761, red color “G’ indicates a SNP) of the methionine import ATP-binding protein (*metN*) gene (Table 1). First, the amplified *metN* gene products of desired six *Salmonella* serotypes were sequenced, and the amplified sequences were aligned. Then we checked SNP positions on the aligned gene sequences. Based on SNPs on the aligned gene sequences, 23 *Salmonella* serotype-specific SNP primers were designed from 13 genes where at least one wild SNP (Appendix A).

### 2.5. Salmonella Serotype-Specific-SNP Primers Design Based on the Appropriate SNP Sites on the Aligned Gene Sequences

The designed 23 sets of SNP-based primers (242–670 bp) were created for confirmation by amplifying the desired *Salmonella* serovars (Figure 1 and Appendix A). One example of the design of the SNP-based primers is shown in Appendix A. SNP-encompassing primer pair ‘16-Sbon’; the forward primer was 5′-ACGGTCTGGGTGAGGTGT-3′ and reverse primer 5′-CCACCGCATCAGAACCGT C-3′. The amplified products with a marker ‘16-Sbon’-amplified *modA* gene of target *Salmonella* were aligned. We observed a few SNP nucleotides on the aligned gene sequences. Based on the SNP sites, five primer sets were developed on the amplified gene (*modA*) sequences (Appendix A). We thus developed the SNP-based marker ‘ModA-1-F/R’, 24-mer forward 5′-ACCCCTGAGATTATCGTTATACTG-3′ and 19-mer reverse primer 5′-ATCGCCCACTGCCAGATGT-3′. In the designed primer, we considered at least two SNPs e.g., forward primer ranges from 130 to 153 (the position of wild SNP ‘G = 153′ and transition mutated SNP site “T = 151; C > T”), and reverse primer ranges from 598 to 616 (the position of wild SNP ‘T = 598’ and the transition mutated SNP site was “C = 600; T > C”). The target amplified product size was 490 bp (Appendix A). A square shape marks the wild SNP and a square shape with black shaded marks the incorporated SNP (Appendix A). Thus, 23 SNP-based markers of *Salmonella* serotype-specific primers were used to amplify the target six *Salmonella* serotype strains. Finally, we confirm the performance of the developed SNP marker with the desired band of target *Salmonella*. The detailed information of six target *Salmonella* gene sequences, their alignment pattern, and designed SNP-based 23 primers were provided in Appendix A.

### 2.6. SNP-Based Multiplex PCR

The amplification with 23 SNP-based markers with target 6 *Salmonella* serovars is time-consuming. Each marker was amplified with all target *Salmonella* strains, a limitation of the developed assay. However, it overcame the expenditure and time for repeated PCR amplification of the SNP-based triplex-marker assay (S1 and S2) (Figure 2 and Table 2). Therefore, we developed a *Salmonella* serotype-specific detection primer set (m-PCR) in a single reaction. In addition, the three primer pairs (SBG-2, ModA-3, SBF-(2)-6) amplified fragment sizes were approximately 498, 373, and 300 bp against *S.* Enteritidis, *S.* Agona, and *S.* Abony, respectively (S1). On the contrary, the three primer pairs (mrcB-1-4, ModA-4, and mrcB-5) amplified fragment sizes were approximately 363, 242, and 637 bp for *S.* Typhimurium, *S. enterica*, and *S.* Typhi, respectively (Figure 2 and Table 2).

### 2.7. Validation of SNP-Based Multiplex Marker with Isolated Salmonella Strains from Wild Animal

For the efficiency test, 21 *Salmonella* were tested with m-PCR, but only 8 *Salmonella* isolates from wild animal feces were identified and evaluated with SNP-m-PCR. The remaining *Salmonella* strains were not identified with m-PCR because these *Salmonellae* were not included with the target six *Salmonella* serotypes. Figure 3 depicts the multiplex PCR marker (S1) generating band drawn to the isolate’s lane no. from 1 to 9, whereas the m-PCR (S2) generating band drawn to the isolate’s lane no. from 10 to 16. The 5 *Salmonella* isolates from leopard cat (*Prionailurus bengalensis*) fecal samples were observed in lanes 5 to 9. These five isolates were well-matched (target band 300 bp) to reference *Salmonella* Abony (BA1800061). Moreover, the *Salmonella* isolates in lanes no.14 and 15 were detected from *P. bengalensis*. The only isolate in lane no.16 was detected from magpie bird (*Pica sericea*) which was well-matched (242 bp) to *Salmonella enterica* subsp. *enterica* NCCP-15756 strain (Figure 3).

## 3. Discussion

*Salmonella*, especially NTS *Salmonella* Typhimurium, Enteritidis are the most prevalent and common serovars [36] for the gastrointestinal disorders resulting from cross-contamination of wild and domestic animal feces [4,37], consumption of animal-originated foods, fresh agricultural produces, i.e., raw fruits and vegetables [38]. In the context of public health, detection of a foodborne outbreak source is essential to remove carrier food items from the market. Molecular investigation and typing of *Salmonella*, the SNP-based typing techniques are essential for identifying the source of a foodborne outbreak. However, *Salmonella* serovars identification methods are based on mainly three basic mechanisms, including restriction fragment analysis of *Salmonella* DNA, PCR amplification of target genes, and SNP-based identification at specific loci in the whole genome sequence [6,20,28]. To date, several molecular approaches have been applied to detect *Salmonellae*, such as PFGE [39], phage typing [40], and multilocus sequence typing (MLST) [41], but they have some limitations. However, SNP-based molecular techniques have recently been proposed as a cost-effective identification method of various bacterial species, including, *Salmonella* [42], *E. coli* [43]; *Mycobacterium* [44]. The SNPs have discrimination power for comparing of bacterial subspecies and at the serovar level using different bioinformatics software [45]. Thus, SNPs could be used as an alternative method to detect outbreaks [2], surveillance of foodborne pathogenic *Salmonella* [46,47], determine models for future outbreaks, and even build an evolutionary and phylogenetic relationship within similar bacterial strains [48,49].

In this study, we searched high-quality SNPs in the coding regions of a respective WGS by comparing each other (reference and compared strains). The quality filtered nucleotide matrix is generated (Appendix A). Furthermore, in this study, 15 primer sets were selected from 13 genes of the *Salmonella* genome-wide searching based on the encompassing SNP sites (Table 1). Primers were designed on the aligned WGS of 13 genes mentioned above (appropriate SNP sites on the aligned genes), approximately 666–863 bp (Table 1). However, only 12 primer sets were amplified with the first and second single-plex PCR, and the rest of the three (4-, 6, and 22-Sbon) primer sets could not produce any band during both PCR cycles (Appendix A). The amplified PCR products were sequenced and aligned for searching suitable SNPs to make serotype-specific primer sets (wild and altered bases of 3′ end of each primer, thus, 23 sets were selected based on aligned sequences). Finally, SNP-containing 6 primers sets from four genes (*fliA*, *modA*, *sopA*, and *mrcB*) were selected for the widely found prevalent six *Salmonella* serotypes detection in an SNP-based m-PCR marker (Table 2). A study used serotype-specific SNPs to identify five *Salmonella* serotypes [42]. Guard et al. [50] postulated that the allele-specific primer was developed based on > 80 SNPs in an adenylate cyclase gene (*cyaA*) of *Salmonella* for the detection of *S. enterica* serovar strains [50]. Roumagnac et al., 2006 conducted a study of approximately 82 SNPs were detected in the partial gene sequences (*n* = 99) of worldwide (*n* = 105) *Salmonella* Typhi isolate, and these SNPs data were used for resolving clear identification of Typhi isolates [51].

This study developed SNP-based primers based on SNPs (wild or mutated transition or transversion) at the 3′ end aligned sequence sites. The introduction of altered bases (transition or transversion) at the end (generally 3′ end) of each primer (except reverse primer SBG-2R) might have changed the codon, which was ultimately used as a target for PCR primer [52,53,54]. This introduction of a mismatched transversion (A-T), or transition (A-G) base pair at the 3′ end sequence could enhance the specific amplification during PCR [55,56]. Thus, transversions (G-C, G-T, A-T, A-C) and transition (A-G, T-C) mutations are required to improve the allele-specific amplification. Based on altered bases (transition and transversions), we developed SNP primers (23 sets, 242–670 bp) for evaluation by amplifying the desired 6 *Salmonella* serovar strains (Figure 1 and Appendix A). In addition, the melting temperature (Tm) generally depends on the GC content of the primer sequences, which is required for PCR conditions adjustment. By incorporating altered bases, the melting temperature of allele-specific SNP-based primers can be fixed to PCR conditions [57,58]. Similarly, the *Salmonella* detected primers were designed based on wild and an altered nucleotide at the 3′ end SNP sites (generally within the three bases) (Table 2 and Appendix A). SNP-based PCR marker was developed for the identification of desire *Salmonella* in a single reaction. An efficient test was conducted with the desired *Salmonella* serovars by PCR amplification of multiplex PCR marker and adjusted to PCR conditions (Appendix A). In a study, the SNP-based phylogenetic analysis of *S*. Enteritidis whole-genome proved that these most prevalent serotypes were clustered in the same lineage, which evolved from the poultry flocs in Brazil [59].

Recently, software algorithms have been used to explore SNP positioning from assembled or raw genome sequences [60]. These new techniques have become increasingly popular for the detection of *Salmonella* compared to other methods. In several investigations, the new technique (SNP-based) has already been applicable in retrospective research studies [52,61,62]. In a study, ten target genes were used to analyze SNPs with common *Salmonella* serovars (Enteritidis, Typhimurium, and Heidelberg). They observed the forty-five nonsynonymous mutations and two most common transition mutations (T ↔ C and A ↔ G), which existed in all *Salmonella* isolates [63]. Similarly, we used 13 genes with two nonsynonymous mutations encompassing primers to sequence all six targets of *Salmonella* serovars for searching SNPs and develop an SNP-based m-PCR marker. Moreover, the most common transition mutations (T ↔ C and A ↔ G) were observed in this study (Table 1 and Appendix A).

Den et al. showed that SNP in WGS is a robust technique compared to multiple-locus variable-number tandem-repeat (VNTR) analysis (MLVA) and pulsed-field gel electrophoresis (PFGE) [64]. In a research, the serovar of *Salmonella* Pullorum was detected by serotype-specific PCR of target gene *rfbS* (paratose synthetase), where a polymorphic site exists at the position of 237, and this SNP-based gene was used to detect and discriminate efficiently between the Pullorum and non-Pullorum [56,58]. Moreover, an allele-specific detection of *S*. Enteritidis was possible based on the SNP site (at position 272 in the plasmid virulence *spvA* gene of *Salmonella*) [65]. Similarly, serotype-specific PCR amplification of four genes (*fliA*; *modA*, *sopA*, and *mrcB*) was used to identify of target six *Salmonella* serotypes, including Enteritidis, Agona, *S. enetrica*, Abony, Typhi, and Typhimurium, respectively, in our study (Table 2). In addition, we observed 870 and 140 SNPs on the whole genome of aligned *S**. enterica* and *S. bogori**,* respectively (Appendix A). SNP encompassing regions of aligned 13 gene sequences were selected for further sequencing with target six *Salmonella*, and SNPs-containing four genes (*fliA*; *modA gene*, *sopA*, and *mrcB*) were validated for the SNP-based multiplex PCR marker. However, PCR enzymes are capable of proofreading activity to correct the mismatch bases [66], but DNA polymerase can increase primers less efficiently (100 to 10,000 folds) compared to matched and mismatched bases [67]. Furthermore, simultaneous incorporation of mismatched bases at the 3′end of each primer (except reverse primer SBG-2R) of third position bases prevented the accurate amplification of targeted *Salmonella* (Table 2). Thus, the detected SNPs could be applied to develop a proper, rapid, alternative, and cost-effective identification method that may contribute to significant improvements in the diagnosis of *Salmonella*.

Species identification with sequence matched to the NCBI, while only sequence cannot identify the *Salmonella* serovar strains due to their high similarity. For serovar identification, we need multiplex PCR. However, our limitation was the developed SNP-based m-PCR, which could not detect all identified *Salmonella* spp. from wild-animal fecal samples beyond the limited serotypes (only six *S. enterica* subsp. enterica serovars). The study’s developed m-PCR (S1 and S2) only detected the six serovars. Furthermore, the proportion of primers mixture was varied owing to possible interference between primers. The S1 and S2 primer sets were separated in this study because there was a difference in the ratio of primers in the primer mixture.

With the developed m-PCR, we should need to verify a number of *Salmonella* serovars from different sources, and further studies should be conducted with accurate serovar determination in the future.

## 4. Materials and Methods

### 4.1. Acquired Salmonella Whole-Genome Sequences (WGS) from GenBank, Searched SNPs, and Designed SNPs-Encompassing Primers

A total of 13 *Salmonella* WGS sequence data, 10 *S. enterica,* and 3 *S. bongori* strains, were obtained from GenBank (ftp://ftp.ncbi.nlm.nih.gov/genomes/refseq/; accessed on 1 August 2022), including reference strain, *S. bongori* NCTC 12419 (NC-015761) and *S. enterica.* serovar Typhimurium str. LT2 (NC-003197) (Appendix A). According to our published article, SNP-based primers and protocols were developed [43]. We used the MUMer package (v3.19, NUCmer algorithm, National Institutes of Health, Bethesda, MD, USA) to investigate the SNP sites on the WGS [61]. SNPs-surrounded primers were developed based on the reference and comparing *Salmonella* strains (Table 1). These primer sets (*n* = 15, SNP encompassing forward and reverse primer sets) were selected based on coding gene sequences, mutation pattern, primer length, G: C content, annealing temperature, the position of the SNPs sites, and so on. In addition, each primer (15 primer sets) was amplified with single plex PCR with the reference target of six *S. enterica* serovars.

### 4.2. Selection and Isolation of Genomic DNA from Serotype-Specific Target Salmonella Serovar Strains

The target six *Salmonella* enterica serovar strains were sequenced by using designed encompassing-primers. For sequencing of target *S. enterica* serovars (*n* = 6), a single plex reaction (20-μL volume) was conducted with the respective primers sets (10 pmol/μL), 10 ng of genomic DNA, 2 μL 10 × buffer, 2 μL dNTP, 0.5 μL (5 unit/μL) Taq DNA polymerase (Qiagen, Hilden, Germany). The amplification reaction was completed in a Bio-Rad T100 thermal cycler (Hercules, CA, USA) programmed with first and second-time PCR cycles consisting of amplification at 95 °C for 5 min, followed by 30–35 cycles of denaturation for 30 s, annealing at 55 °C for 30 s (first PCR), and annealing at 50 °C for 30 s (second PCR), polymerization at 72 °C for 1 min 30 s, and final elongation at 72 °C for 10 min. After amplification, 5 μL of each PCR product was analyzed on a 1.5% (*w*/*v*) agarose gel. The amplified PCR products were purified (Gel & PCR Purification Kit; Biomedic Co., Ltd., Seoul, Korea) and sequenced using a BigDye Terminator v3.1 Cycle Sequencing Kit (Applied Biosystems, Foster City, CA, USA) and ABI 3730 DNA Analyzer (Applied Biosystems, Foster City, CA, USA).

We carefully checked one or more SNP positions on the multiple aligned gene sequences of each target reference *Salmonella* (Appendix A). The designed primers were amplified efficiently at the target band with first and second PCR cycles. Furthermore, if SNP-based encompassing primers were produced ambiguous, overlapping/non-target peaks, then they were removed for further analysis. *Salmonella* serovar-specific-SNP markers were designed to discriminate single base changes through experimental optimization [30,32,33]. However, *Salmonella* serotype-specific SNP primers were designed on the aligned gene sequences where at least one wild-type and one altered SNP were present in each primer set. Furthermore, if SNP-based encompassing primers produced ambiguous, overlap-ping/non-target peaks, they were removed for further analysis. *Salmonella* serovar-specific-SNP markers were designed to distinguish any base changes (SNP) by the experimental optimization [30,32,33].

All the 13 different genes in a *Salmonella* genome such as conserved hypothetical protein (*SBG*); dedA family integral membrane (*dedA*); conserved hypothetical protein (*yacG*); penicillin-binding protein (*mrcB*); tRNA(Ile)-lysidine synthase (*mesJ*); methionine import ATP-binding protein (*metN*); pyrimidine-specific ribonucleoside hydrolase (*rihA*, *rihB*); molybdate-binding periplasmic protein (*modA*); formimidoylglutamase (*hutG*); hypothetical ABC transporter ATP-binding (*yehX*); pyruvate formate-lyase 3-activating enzyme (*ybiY*); molybdopterin biosynthesis MoeB protein (*moeB*); candidate type three secretion system effector protein (*sopA*) including SNP sites, were amplified using the target six *Salmonella* serovars. The amplified PCR products of the target *Salmonella* were re-sequenced and aligned using BioEdit sequence alignment editor, version 7.0.0 (Tom Hall, North Carolina State University, United States). The *Salmonella* detected primers were designed based on a wild and an altered nucleotide at the 3′ end SNP sites (transition/transversion mutation). Thus, all the designed primers were further analyzed using NetPrimer (https://www.premierbiosoft.com/netprimer/ accessed on 1 August 2022) to choose the best primer pairs.

### 4.3. Salmonella Serotype-Specific SNP-Based Multiplex-PCR Marker

The SNP-based serotype-specific primers were designed to identify the target *Salmonella*. However, the amplification of all six target *Salmonella* serotypes with all 23 SNP-based primers is time-consuming. To overcome the limitation of repeated PCR amplification, we developed an SNP-based multiplex PCR kit to identify target six *Salmonella* serovars in a single reaction.

Multiplex (S1 and S2) PCR indicates the difference in PCR mixture ratio (mixture of three primers sets), which were conducted with similar PCR conditions (annealing reaction at 60 °C for 30 s, 30 PCR cycles, see below). The adjusted concentration of the primer mixture of S1 (three primer sets, SBG-2F/R, ModA-3-F/R, and SBG(2)-6F/R) was 1:1:1 while the primer mixture of S2 (mrcB-1-F/R, modA-4-F/R, and mrcB-5-F/R) was 1:3:3, respectively. From a mixture of each forward and reverse primer, only 3 μL PCR mixture (10 pmol/μL) was used in a final volume of 20 μL. A multiplex reaction (20-μL volume) was conducted with the respective primer’s mixture 3 μL (10 pmol/μL), 5 ng of genomic DNA (5 ng/ul), 10 μL Hot Start *Taq* master mix including Hot Start *Taq* DNA Polymerase, dNTPs, MgCl2, KCl, and stabilizers (Takara Bio Inc., United States), and PCR grade water 6 μL. The multiplex PCR reaction was conducted for 5 min at 95 °C, followed by denaturation 95 °C for 30 s, annealing reaction at 60 °C for 30 s, extension at 72 °C for 30 s, and final extension at 72 °C for 5 min and holding temperature at 4 °C for an unlimited period. The amplification reaction was completed in a Bio-Rad T100 (Hercules, CA, USA).

### 4.4. Validation of SNP-Based Multiplex Marker with Reference Strains and Laboratory Isolated Salmonella Strains

The isolated laboratory *Salmonella* from wild-animal fecal samples and reference bacteria strains were tested with multiplex PCR marker for efficiency tests. The cross-reaction was observed with the reference *Salmonella* strains (*n* = 6) and the laboratory *Salmonella* isolates (*n* = 21) from wild-animal and bird feces. Wild animal and bird fecal (*N* = 699) samples were collected from various agricultural regions and mountainous areas over three years (2015–2017) across South Korea (unpublished data). From these wild-animal fecal samples, 21 *Salmonella*-positive fecal based on traditional cultural, biochemical, serological, and molecular approaches according to the methods described earlier [68,69,70]. In a cultural process, a non-selective buffer peptone water (BPW) was used for primary enrichment of *Salmonella* at 37 °C for 18–24 h. One milliliter (1000 µL) of primary enrichment broth was added to 9 mL of Muller Kauffmann Tetrathionate enrichment broth (Difco, Becton Dickinson, NJ, USA) at 40–44 °C for 18–24 h, while 100 µL of primary enrichment broth culture was incubated in 10 mL Rappaport-Vassiliadis (RV) enrichment broth (Oxoid, UK) at 37 °C for 18–24 h. After primary and secondary enrichment, 10 µL of each sample was streaked on *Salmonella*-selective, *Salmonella*-Shigella (SS) and Hektoen enteric (HE) agar media (Difco, Becton Dickinson, United States), and incubated at 37 °C for 24–48 h. A presumptive *Salmonella* colony from SS and HE agar media was identified by amplifying the *invA* and *iroB* primer sets, the genes (invasion gene, *invA*, and iron-regulated virulence gene, *iroB*), which are shared by all *Salmonella* species (35, 36). The identified *Salmonella* isolates from wild animal feces were tested and evaluated by the developed SNP-based m-PCR primers in this study. In addition, sensitivity tests were also conducted with Gram-negative and Gram-positive bacteria (data not shown). We provided the image of gram negative six *Salmonella* serovar strains.

Nevertheless, the sample collections were conducted under permission and the guideline of the local government. In addition, the protocol of this experiment was permitted by the Institutional Animal Care and Use Committee of Kangwon National University, Chuncheon, Korea. The approval number was KW210701-1. Moreover, the obtained *Salmonella* spp. sequences were compared with similar deposited sequences in NCBI, BLASTN2.2.31+ [71] analysis. The genetic sequence data (21 isolate-sequences and 6 reference sequences) were deposited into the NCBI with the accession numbers (OM793284-OM793310).

## 5. Conclusions

So far, we know few SNP-based multiplex PCR detection methods in which widely prevalent six *S. enterica* serovars were detected in a single reaction. In this stud, the developed m-PCR test could be applied to investigate and distinguish serovars in a single PCR tube. The newly developed m-PCR marker is a novel, simple and reliable method for the identification of widely found six *S. enterica* subsp. *enterica* serovar strains (typhoidal and nontyphoidal *Salmonella*). However, to ensure the high specificity of developed mPCR, further analysis should be conducted with a number of sample sources *Salmonella* serovars and other bacterial strains (wild animals, foods, food animals, environmental, and clinical samples).

## 6. Patents

Yung Chul Park, M.M. Rahman, and S.J. Lim. Development of multiplex PCR kit and detection of *Salmonella*. Kangwon National University, Korea. Patent No: 10-182857.

## Figures and Tables

**Figure 1 pathogens-11-01075-f001:**
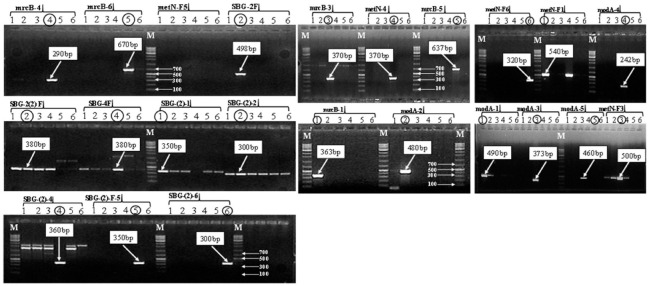
A Figure depicts the PCR amplification of six target *Salmonella* serovars with 23 SNP-based primers. ‘M’ denotes DNA 100 bp marker. The gel lane numbers were presented in each section (1–6): lane No.1 = *Salmonella enterica* subspecies enterica serovar Typhimurium, *S.* Typhimurium, (NCCP-14760); No.2 = *S*. Enteritidis (NCCP-14545); No.3 = *S.* Agona (NCCP-12231); No.4 = *S. enterica* (NCCP-15756); No.5 = *S*. Typhi (NCCP-14641); No.6 = *S.* Abony (BA1800061). Detailed primer sequence information is provided in Appendix A. Here, all the primers were presented on top of the gel image, the target length of each primer was provided in a white shaded box, and the desired band of each primer of target *Salmonella* was marked by a black circle. We selected only the single intense band (primer, mrcB-1) with the SNP-based single-plex multiplication for further analysis. The primers produce a non-target light band (SBG-4F/R), a double band (SBG (2)-4); and multiple bands [(SBG-2(2); SBG (2)-2; SBG (2)-1)]; which were not considered for the identification of target *Salmonella* serovars.

**Figure 2 pathogens-11-01075-f002:**
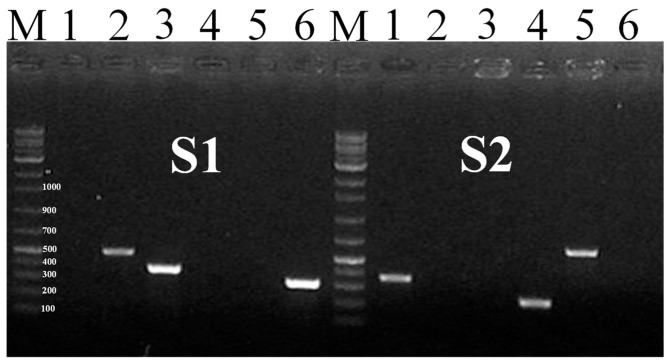
The PCR amplification of 6 target *Salmonella* serovars with SNP-based multiplex serotype-specific *Salmonella* primer set (S1 indicate the three distinguish band of 2, 3, and 6 no lane; and S2 indicate the rest of three distinguish band of 1, 4, 5 no lane for clear visualization with the necked eye). The band ‘M’ denotes the DNA 100 bp marker. The lane numbers were presented in each section (1–6): the gel lane no = *Salmonella*
*enterica* subspecies *enterica* serovar (*Typhimurium*) name (reference no, target length) of S1: lane No.2 = *S.* Enteritidis (NCCP-14545; 498 bp); No.3 = *S.* Agona (NCCP-12231, 373 bp); No.6 = *S.* Abony (BA1800061, 300 bp) and S2: lane No.1 = *S.* Typhi (NCCP-14641, 363 bp); No.4 = *S. enterica* (NCCP-15756, 242 bp); No.5 = *S.* Typhimurium (NCCP-14760, 637 bp).

**Figure 3 pathogens-11-01075-f003:**
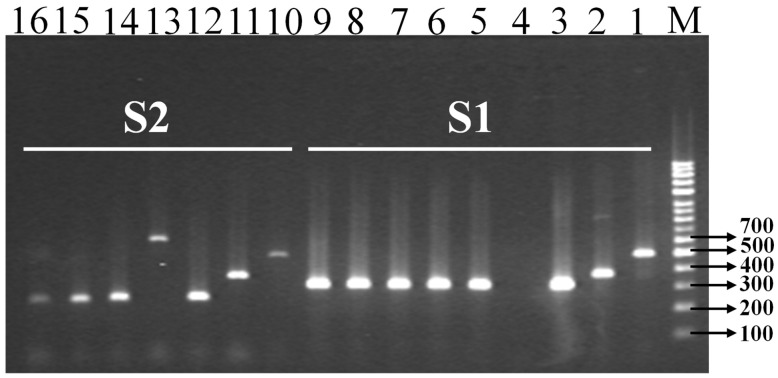
PCR amplification of *Salmonella* serovar-specific multiplex primer set (m-PCR) with our lab isolated *Salmonella* and reference *Salmonella*. The ‘M’ denotes DNA 100 bp marker. *Salmonella enterica* subspecies *enterica* serovars (*S*. serovar); the gel lane no. (1–9) for m-PCR primer sets, S1: three reference *Salmonella* (lane, 1–3) lane No.1 = *S.* Enteritidis (NCCP-14545); No.2 = *S.* Agona (NCCP-12231); No.3 = *S.* Abony (BA1800061) and No.4 = negative control (only PCR mixture without DNA); No.5 = *Prionailurus bengalensis*; No.6 = *P. bengalensis*; No.7 = *P. bengalensis*; No.8 = *P. bengalensis*; No.9 = *P. bengalensis* and the gel lane no. (10–16) for m-PCR primer sets, S2: three reference *Salmonella* (lane, 10–12); lane No.10 = *S.* Typhimurium (NCCP-14760); No.11 = *S.* Typhi (NCCP-14641); No.12 = *Salmonella enterica* subsp. *enterica* (NCCP-15756); No.13 = reference non-*Salmonella*, *Enterobacter cloacae* (NCCP-14621); No.14 = *P. bengalensis*; No.15 = *P. bengalensis*; and No.16 = *Pica sericea*.

**Table 1 pathogens-11-01075-t001:** The developed single nucleotide polymorphisms (SNPs)-encompassing primers information based on whole-genome sequences (WGS) of *Salmonella bongori*.

No. ^c^	Forward Primer (5′-3′)	Reverse Primer (5′-3′)	Amplicon Size (bp)	Gene (Position of SNP: Nucleotide), Flanking Sequences in between Ambiguous Code ^b^	One-Letter Amino Acid (a.a) Code of Comparing *Salmonella bongori* Strains *RKS3044/N268-08* -a.a Position of Ref. *S. bongori* NCTC 12419-a.a Code (Mutation Types) ^a^	Respective Genes
01	GGGGAAATGTTGGCGGGA	TTATGCCCGGTGCCATGG	735	*SBG_*RS00105 (20978: G) CAACCTGCCDACCCCGATGAG	K/Y-145-D (nonsynonymous)	Conserved hypothetical protein (*SBG*)
04	TTGCTGGTCGCCTTCCTG	CGTATCGCGTGGCAAGGA	863	*dedA* (104317: G) TCTGGCTGGHGCCGCTATTGA	G/G-163-G (Synonymous)	DedA family integral membrane (*dedA*)
06	CGCGTGATGGAGCAGGAT	CCTCACACAGGCGCTGAA	674	*yacG* (146357: T) CCAGACGACBGCTTTACCACA	A/P-15-A (Synonymous)	Conserved hypothetical protein (*yacG*)
09	GGCGTTGAAGAAGCAGCG	ACGGCCTACCCAGGTGAT	799	*mrcB* (210347: C) CGCCAGCGGBGGAAATCGCGC.	G/G-626-G (Synonymous)	Penicillin Binding protein (*mrcB)*
11	TTCTGGCCAGCGACCTTG	TGCCAGTTTCAGCCACCC	714	*mesJ* (263017: G) TGAACTGCGBCAACCGCGCGC.	R/R-349-R (Synonymous)	tRNA (Ile)-lysidine synthase (*mesJ*)
12	ATTGGCACGCTGTCAGCT	TGCCGGTAAAAGCACGCT	681	*metN* (270836: G) CGGATCGAGGBCGCTGGTCGC.	A/A-169-A(Synonymous)	Methionine import ATP-binding protein (*metN*)
13	GCTGTACCTGCCGACTGG	GTTCCCCACGGGCTATGG	797	*rihB* (576128: T) GCGTATGACDCTGCAGTACG.	T/T-69-T (Synonymous)	Pyrimidine-specific ribonucleoside hydrolase (*rihB*)
14	TCCCCTGTGTTTCGACGC	ACGCCGGATAAGACGCTG	682	*rihA* (626519: G) CTCGGCAGCBGCGTCCAGTT.	P/P-167-P (Synonymous)	Pyrimidine-specific ribonucleoside hydrolase(*rihA*)
15	GCGGGAAACTCCTGTGCT	CAACACCCGGCAGCAAAC	766	*modA* (683698: C) ACTACACCGVCGCTTCATGG.	R/R-104-R (Synonymous)	Molybdate-binding periplasmic protein (*modA*)
16	ACGGTCTGGGTGAGGTGT	CCACCGCATCAGAACCGT	836	*modA* (747388: T) TGCGGCGGADTATAAAAAAGA.	D/D-47-D (Synonymous)	Molybdate-binding periplasmic protein (*modA*)
18	GCATCTGGATCTGCGCCA	TCGGCGACAAAGGTTCCC	751	*hutG* (766287: G) AATGCCGGCBTTTCCGCCCC.	A/A-252-A (Synonymous)	Formimidoylglutamase (*hutG*)
19	TCACGGCGGGTAAGAGGA	ATGAGATTCGCCAGGCCG	666	*yehX* (792736: T) GGCTTTGCCDAGCTGACTTT.	S/S-397-S (Synonymous)	Hypothetical ABC transporter ATP-binding (*yehX*)
21	CTGCTTAAACGGCGCGTC	TGGTGCGGCATGATCCTG	738	*ybiY* (827585: T) TGAGCCAGGHTGGAAAAATGG	N/Y-87-S (Nonsynonymous)	pyruvate formate-lyase 3-activating enzyme (*ybiY*)
22	CCGAACAGACGGCTCAGG	CCGGACATCAAGGGTCGC	681	*moeB* (828767: C) GACGCCTGCVCCGGCCAGATA	G/G-52-G (Synonymous)	Molybdopterin biosynthesis MoeB protein (*moeB*)
24	GTAGTGTGGCGGGCTGAG	CTGGTAAGCGTGCTGGCC	801	*sopA* (1032435: T) CTCATAAAGHGCCGCGGCTTT	A/A-494-A (Synonymous)	Candidate type three secretion system effector protein (*sopA*)

^a^ Reference genome of *Salmonella bongori* str. NCTC 12419 (NC_015761), the comparing strains (*S. bongori* serovar 48:z41:-str. RKS3044 (NZ_CP006692), and *S. bongori* N268-08 (NC_021870) and ‘1-letter’ amino acid codes, K = Lysine, Y = Tyrosine, G = Glycine, A = Alanine, R = Arginine, T = Threonine, P = Proline, D = Aspartate, S = Serine, N = Asparagine; ^b^ Ambiguous codes indicate D = A/G/T; B = C/G/T; V = A/C/G; H = A/C/T; ^c^ indicates primer code no.-(such as ‘01-Sbon’, ‘04-Sbon’, ‘06-Sbon’and so on, a total 15 primer sets) which acquired based on the suitable primers among multiple primers generating by bioinformatics software and we considered the selected primer indicators i.e., amplicon length, G: C content, synonymous and nonsynonymous amino acid mutation in a protein-coding gene sequence, annealing temperature, the position of the SNPs sites, and so on. SNPs sites are marked by bold International Union of Pure and Applied Chemistry (IUPAC) codes in flanking sequence (D = A/G/T; B = C/G/T; V = A/C/G; H = A/C/T).

**Table 2 pathogens-11-01075-t002:** The developed multiplex primers based on single nucleotide polymorphisms (SNPs) of the whole genome of *Salmonella*.

Gene	Primer	Sequence (5′-3′) ^#^	(Mer bp)	Size (bp)	*Salmonella* Serovar Strains	m-PCR
RNA polymerase sigma factor FliA (*fliA*)	SBG-2F	TTACCAGGAAGAGCTCGAC	19	498	*Salmonella* Enteritidis	S1
	SBG-2R	CGGTGCCATGGCTCATCTCG	20			
Molybdate-binding periplasmic protein (*modA*)	ModA-3-F	TCGCAGGGGCGACATTATCTTCCA	24	373	*Salmonella* Agona	
	ModA-3-R	AGACGAATCCAGTCCGTTTTGCTA	24			
E3 ubiquitin-protein ligase sopA (*sopA*)	SBG (2)-6F	GCTGGTTCAGCTCCCCATTA	20	300	*Salmonella* Abony	
	SBG (2)-6R	CGGACTGGACAACCCGCTCC	20			
Penicillin binding protein (*mrcB*)	mrcB-1-F	TGGCGTTAGGTCTACCGTCA	20	363	*Salmonella* Typhi	S2
	mrcB-1-R	TTGTCGTCCCGGTTTTATCG	20			
Molybdate-binding periplasmic protein (*modA*)	ModA-4-F	TTACGCCTGGTCGCAGGGACA	21	242	*Salmonella* *enterica*	
	ModA-4-R	CATTTCTGATCAGCAGAGATGGAG	24			
Penicillin binding protein (*mrcB*)	mrcB-5-F	GGCGGAGCCGCAGTATACT	19	637	*Salmonella* Typhimurium	
	mrcB-5-R	TGTCGTCCCGGTTTTACTCA	20			

^#^ Red color indicates the natural SNP and blue color indicates the alterd/artificial mutated nucleotide (transition and transversion).

## Data Availability

The genetic sequence data used were deposited into the following GenBank accession numbers (OM793284-OM793310). The data will be available on request to the corresponding author.

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
