# Peer review of "Genome-Wide Searching Single Nucleotide-Polymorphisms (SNPs) and SNPs-Targeting a Multiplex Primer for Identification of Common Salmonella Serotypes"

_pathogens, 2022, doi:10.3390/pathogens11101075_

Round 1

Reviewer 1 Report

This work developed a multiplex PCR method for species identification of six common Salmonella serotypes. The serovar-specific SNP markers were selected by aligning 13 Salmonella WGS sequence data from GenBank. Six primer pairs designed from four functional genes were to target the six Salmonella serotypes, and single plex PCR and multiplex PCR were constructed. The multiplex method can be applied for rapid and reliable detection of prevalent Salmonella serotypes detection in the environment, animal originated foods, water, and animals. The aim of the manuscript is interesting; however, the results section should be more concise.

Some detailed comments below:

1.     The purpose of this manuscript is to “identify novel sensitive and reliable serovar-specific targets” mentioned in line 66, while there is no sensitivity result for this method in the manuscript. And the low sensitivity of the detection may cause the negative results. The sensitivity of this method should be added for reliable results.

2.     There is a lack of explanation of why the six primer pairs for multiplex PCR into two primers sets (named S1 and S2 in line 179) instead of six primer pairs in one PCR tube.

3.     In the validation results of fecal samples from the wild animals, only 8 Salmonella isolates of 21 were detected by the multiplex methods. How to determine the positive results to the target six Salmonella serotypes but not the rested Salmonella strains were not identified before. It’s better to added the sequencing alignment results of the PCR products. 

4.     The reference styles were not unified in the whole manuscript, such as line 243,244,374. Please check manuscript format of the journal and revise carefully.

Author Response

Title

Genome-wide Searching Single Nucleotide-Polymorphisms (SNPs) and SNPs-targeting a Multiplex Primer for Identification of Common Salmonella Serotypes

pathogens-1843946

Reviewer: 1

Comments: 1.

The purpose of this manuscript is to “identify novel sensitive and reliable serovar-specific targets” mentioned in line 66, while there is no sensitivity result for this method in the manuscript. And the low sensitivity of the detection may cause the negative results. The sensitivity of this method should be added for reliable results.

Response 1:

Thanks for your real observation and nice comment. We agree with you. The sensitivity results with the other bacterial strains were checked but no data picture was provided.

Each PCR primer was checked by all target six Salmonella serovars that proved the specificity and sensitivity tests were also conducted with related Gram-negative and Gram-positive bacteria (data not shown).

We need to make ensure further analysis of a number of Salmonella serovars collected from variety sample sources (not only animal sources, but also other sources, including foods, food animals, environmental, and clinical)

Comments: 2.

There is a lack of explanation of why the six primer pairs for multiplex PCR into two primers sets (named S1 and S2 in line 179) instead of six primer pairs in one PCR tube.

Response 2:

Thanks for your nice comment. The six primers already were conducted in a reaction but PCR image should make clear the band distinction from each other.

When detecting each serovar in a genomic DNA mixture of six serovar gDNAs, the detection sensitivity of each set of primers in the primer mixture volume was significantly affected (please refer to our preliminary PCR photo data). Finally, it was possible to secure a stable sensitivity at a ratio of 1:3:3 at 60 degrees (mrcB-1-F/R, modA-4-F/R, and mrcB-5-F/R). Therefore, the proportion of primers was very important for detection sensitivity. It may be reason for interference between primers. If S1 and S2 are mixed, the S1 and S2 primer sets are separated in this study because it is expected that the sensitivity will decrease due to interference between primers because there is a difference in the ratio of primers in the primer mix.

Basically the S1 multiplex PCR primer mixture were ratio of the (SBG-2F/R, Mod-3-F/R and SBG(2)-6F/R, PCR primer mixture ratio 1:1:1, respectively; each of PCR was taken only 1 ul), and finally PCR mixture was taken 3 ul for final PCR- mixture for  PCR running, and the S2 primer mixture consist of mrcB-1-F/R, ModA-4-F/R, and mrcB-5-F/R ratio was 1:3:3, respectively and run the PCR mixture with target samples. Thus, we differentiate each band length of each amplified PCR image. However, we should conduct more sample analysis collected from diverse sources, including clinical, food animal, and nvironmental samples for validation of designed primers. 

We added the following text lines from 399-412  “ Multiplex (S1 and S2) PCR indicates the difference of PCR mixture ratio (mixture of three primers sets) which were conducted with similar PCR condition (annealing reaction at 60oC for 30s, 30 cycles). The adjusted ratio of primer mixture of S1 (three primer sets, SBG-2F/R, ModA-3-F/R, and SBG(2)-6F/R) was 1:1:1 while primer mixture ratio of S2 (mrcB-1-F/R, modA-4-F/R, and mrcB-5-F/R) was 1:3:3, respectively. From the mixture of each group of S1 and S2, only 3 μl PCR mixture (10 pmol/μl) was used in a final volume 20 μl. A multiplex reaction (20-μl volume) was conducted with the respective primer’s mixture 3 μl (10 pmol/μl), 5 ng of genomic DNA (5 ng/ ul), 10 μl Hot Start Taq master mix including Hot Start Taq DNA Polymerase, dNTPs, MgCl2, KCl and stabilizers (Takara Bio Inc.), and PCR grade water 6 μl. The multiplex PCR reaction was conducted for 5 minutes at 95oC, followed by denaturation 95oC for 30s, annealing reaction at 60oC for 30s, extension at 72oC for 30s, and final extension at 72oC for 5 minutes and holding temperature at 4oC for unlimited period. The amplification reaction was completed in a Bio-Rad T100 (Hercules, California, USA).” After amplification, 5 μl of each PCR product was analyzed on a 1.5% (wt/vol) agarose gel. The amplified primer products were purified (Gel & PCR Purification Kit; Biomedic Co., Ltd., Seoul, Korea) and sequenced using a BigDye Terminator v3.1 Cycle Sequencing Kit (Ap-360 plied Biosystems, Foster City, CA, USA) and ABI 3730 DNA Analyzer (Applied Biosystems, Foster City, CA, USA).

For reviewer’s kind observation the following PCR images of different ratio of S2 primer mixture are provided:

Comments: 3.  In the validation results of fecal samples from the wild animals, only 8 Salmonella isolates of 21 were detected by the multiplex methods. How to determine the positive results to the target six Salmonella serotypes but not the rested Salmonella strains were not identified before. It’s better to added the sequencing alignment results of the PCR products. 

Response 3:

The rest of Salmonella were identified with sequencing and serological testing. We identified 21 Salmonella-positive wild animals fecal, amplified and sequenced and submitted to the NCBI following accession number: OM793284-OM793304. The eight Salmonella were identified with mPCR and the rest of the Salmonella (13 Salmonella strains) were determined serological and sequencing similarity (they are the included others serovars including S. enterica subsp. enterica serovar Anatum, S. enterica subsp. diarizonae str.) (unpublished data).

We compare our sequencing result of six reference Salmonella enterica serovar strains, including Salmonella enterica subsp. enterica serovar Abony (S. Abony, accession no. OM793305), and S. Typhimurium (accession no. OM793306), S. Enteritidis (accession no. OM793307), and S. Agona (accession no. OM793308), S. enterica subsp. enterica (accession no. OM793309), and S. Typhi (accession no. OM793310). The genetic sequences data (21 isolate-sequences and 6 reference sequences) were deposited into the NCBI with the GenBank accession numbers (OM793284-OM793310).

Comments: 4. The reference styles were not unified in the whole manuscript, such as line 243,244,374. Please check manuscript format of the journal and revise carefully.

Response 4: Fixed, according to Journal format.

Reviewer 2 Report

This research paper entitled “Genome-wide Searching Single Nucleotide Polymorphisms (SNPs) and SNPs-targeting a Multiplex Primer for Identification of Common Salmonella Serotypes” is much worthy of investigation. Target SNP-sites were screened to amplify the target Salmonella serovar strains. Overall paper is reporting some meaningful results. But I have some minor concerns regarding this manuscript.
Q:  Why choose those six target Salmonella serotypes?

Are they representative of all pathogenic Salmonella identified on clinic?

It is recommended that the author make a more detailed and reasonable explanation in the manuscript.

Line 327: Moreover, does the author have any idea to remedy that mentioned limitation?

The manuscript should be further improved by English editing.

1. It seems NC-015761 was a wrong accession number.

2. In Table 2, what's the meaning of blue and red color base?

Author Response

Reviewer 2:

Comments: 1: Why choose those six target Salmonella serotypes?

Response 1: The six-target Salmonella are including typhoidal (S. typhi) and nontyphoidal (the major prevalence S. Typhimurim, Enteritidis are the most prevalence serotypes in the world). In European food safety authority, salmonellosis is the most reported (2nd) cases of zoonotic diseases in EU. (see the link: https://www.efsa.europa.eu/en/topics/topic/salmonella). In the United States approximately, 1.4 million NTS Salmonella infections and over 400 deaths happened in a year. Even in sub-Saharan Africa, these two serovars have been reported to account for 59–95% of all bacteremic non-typhoidal Salmonella infections (Kariuki et al 2006, see the link https://doi.org/10.1186/1471-2180-6-101)

Comments: 2: Are they representative of all pathogenic Salmonella identified on clinic?

Response 2: The most of the representative Salmonella are identified on clinic. Salmonella enterica Salmonella Typhi causes human diseases which are host specific. Animals (domestic and wild) are the zoonotic carriers and cause infections to human. Some NTS are more prevalence (S. Typhimurim and Enteritidis) compare to others (S. Agona and Abony). The representative NTS are the causative agents of invasive diseases (bacteremia and focal infections such as meningitis) in infants, young children, and immunocompromised patients.

Comments: 3: Line 327: Moreover, does the author have any idea to remedy that mentioned limitation?

Response 3: According to CDC, Salmonella contain two major species, S. enterica and S. bongori. S. enterica has five major subspecies. Among them, Salmonella enterica subsp. enterica contain approximately 1500 serovars. Interestingly all serovars are not equal to infectious to human.  Sometimes new Salmonella serovars are evolving from diverse sources. Therefore, in practically Salmonella serovars (more than 2500 serovars) hard to distinguish but the limitation can be overcome in the way of investigating the top most serovars by using an improved method (SNP-based new detection method based on large scale whole genome sequence). For instance: Pan-genome Analyses used for wider scale identification of Salmonella Species, Subspecies, and Serovar, see the link: https://doi.org/10.3389/fmicb.2017.01345  

Comments: 4: The manuscript should be further improved by English editing.

Response 4: The manuscript was corrected by the native speaker

Comments: 5: It seems NC-015761 was a wrong accession number.

Response 5: We checked carefully in NCBI, it is correct and shared a snap image of searching result

  1.  

Comments: 6: In Table 2, what's the meaning of blue and red color base?

Response 6: Fixed

Round 2

Reviewer 1 Report

The authors have revised the majority of problems we proposed, while the sensitivity test we referred to is the minimum detection limit of their primers, while the yes or no detection may conclude to the specificity but not the sensitivity in Table S5. If the sensitivity test can not be provided, it’s better to use high specificity instead of sensitivity in the conclusion section.

The manuscript can be considered for publication after the above problem addressed.

Author Response

Reviewer: 1

Comment: 1.

The authors have revised the majority of problems we proposed, while the sensitivity test we referred to is the minimum detection limit of their primers, while the yes or no detection may conclude to the specificity but not the sensitivity in Table S5. If the sensitivity test can not be provided, it’s better to use high specificity instead of sensitivity in the conclusion section.

The manuscript can be considered for publication after the above problem addressed.

Response 1:

Again, thanks for your nice suggestion and proposal for acceptance after correction. We mentioned the minimum detection limit of primers as sensitivity. The sensitivity results with the other bacterial strains were not provided (truly negative). In Table S5, we provided only the specificity test with only the six positive Salmonella serovar strains with 23 sets of designed primers. As per your recommendation, we changed the sentence structure with high specificity instead of sensitivity in the conclusion section.